# Measuring Knowledge of Healthcare Providers on Pediatric Palliative Care with an Online Questionnaire Based on the National Core Curriculum in Italy

**DOI:** 10.3390/healthcare11131971

**Published:** 2023-07-07

**Authors:** Elisa Zanello, Roberta Vecchi, Giulia Zamagni, Maria Celeste Biagi, Irene Bruno, Elisa Cragnolin, Elisabetta Danielli, Silvia Paoletti, Marco Rabusin, Luca Ronfani, Emanuelle Pessa Valente

**Affiliations:** 1Institute for Maternal and Child Health IRCSS “Burlo Garofolo”, 34137 Trieste, Italy; elisa.zanello@burlo.trieste.it (E.Z.); mariaceleste.biagi@burlo.trieste.it (M.C.B.); irene.bruno@burlo.trieste.it (I.B.); elisa.cragnolin@burlo.trieste.it (E.C.); elisabetta.danielli@burlo.trieste.it (E.D.); marco.rabusin@burlo.trieste.it (M.R.); luca.ronfani@burlo.trieste.it (L.R.); emanuelle.pessavalente@burlo.trieste.it (E.P.V.); 2Pineta del Carso Hospice, 34011 Trieste, Italy; r.vecchi@polifvg.it; 3ANVOLT Trieste—Associazione Nazionale Volontari Lotta Contro i Tumori, 34135 Trieste, Italy; silvia.paoletti1227@gmail.com

**Keywords:** pediatric palliative care, pain therapy, education, training, questionnaire development

## Abstract

There is a lack of highly reliable tools evaluating healthcare professionals’ competences on Pediatric Palliative Care (PPC) and Pain Therapy (PT). The aim of this study is to document the development of an online questionnaire to assess Perceived, Wished and Actual Knowledge of healthcare workers on PPC/PT. The tool was built on the basis of the Italian Society for Palliative Care PPC Core Curriculum (CC) for physicians, nurses and psychologists. Face validity, internal consistency and the underlying structure were evaluated after a field testing in a referral hospital, Friuli-Venezia Giulia, Italy. One hundred five respondents completed the questionnaire. High internal consistency for both scales of Perceived and Wished Knowledge was found (α = 0.95 and α = 0.94, respectively). Psychologists reported higher levels of self-Perceived skills on the psychosocial needs of the child and family at the end of life (*p* = 0.006), mourning (*p* = 0.003) and ethics and deontology in PT/PC (*p* = 0.049). Moreover, when Actual Knowledge was tested, they also provided the highest number of correct answers (*p* = 0.022). No differences were found by profession for Wished Knowledge. The questionnaire showed promising psychometric properties. Our findings suggest the need of continuous training in this field and identify contents to be addressed in future training programs.

## 1. Introduction

In recent decades, the number of children with special healthcare needs (CWSHCN) is growing as a result of the scientific and technological improvements in healthcare [1]. A few number of CWSHCN have life-threatening and life-limiting conditions requiring palliative care (PC). The conditions eligible for PC in newborn, infant, child, or adolescent patients include a broad and heterogeneous range of diseases, with specific needs related to the child’s continuous development [2]. According to the most recent estimates, more than 21 million children annually worldwide need a palliative approach and of them, more than 30% require specialized Pediatric Palliative Care (PPC) [3]. In Italy, et al. estimate that 34–54 children per 100,000 inhabitants (of any age) require PPC, of which 18 require specialized PPC [4]. However, there is a lack of knowledge among healthcare workers about palliative care and training is often not appropriate, even in high income countries [5,6,7,8,9,10]. To ensure appropriate and high-quality care for patients and families, specific education and training must be provided to all professionals involved in healthcare sectors in all settings [11]. In recent years, Italy has issued regulations to protect the citizens’ right to access PC and Pain Therapy (PT) and introduced the recognition of the specialist discipline in PC and educational paths for health workers training both at university and professional level [12,13]. Like other scientific societies worldwide, the Italian Society of Palliative Care defined a “Core Curriculum for Pain Therapy and Pediatric Palliative Care—Team Work” on PT and PPC for three health professional profiles, physicians, nurses and psychologists, for simplicity called here Core Curriculum (CC) [14,15]. The development of the CC was motivated by the need for an accurate definition of the professionals’ tasks and the required education. For this reason, it provides for each professional profile a detailed synthesis of those competences deemed essential to work in the field of PC. In particular, there are three levels of professional exposure to PC: basic (i.e., students and qualified professionals occasionally dealing with situations requiring a palliative approach), intermediate (i.e., qualified workers daily involved in the PC setting, as oncology or geriatrics) and advanced (i.e., heads of PC units or consultants).

Therefore, the utilization of CC aims to ensure the adequacy of the provided care, and also a qualified and uniform response throughout the country to all children requiring PC and PT and their families. However, the inadequate training of healthcare professionals in PPC still represents an important barrier to the full implementation of PPC in all regions of the country and comprehensive information on the PPC education and its dissemination is still lacking. In fact, as showed by the survey conducted by Benini et al., among pediatric residents in Italy, 91.4% of the respondents consider PPC an essential competence for pediatricians, even though only 37% attended a PPC service during their postgraduate training programs [16].

Several surveys have comprehensively investigated educational needs of healthcare providers on PPC focusing on individual professions, periods of life or specific domains [7,17,18,19]. However, to our knowledge, there is a lack of validated tools investigating the healthcare professionals’ competence on PPC considering both subjective (self-evaluation) and objective (evaluation by others) criteria, with a multi-professional approach. In particular, using innovative methods enables rapid data collection (online surveys) in high income countries. Our study aimed to document the development of a questionnaire to assess the self-perceived knowledge (Perceived Knowledge), the topics to explore further (Wished Knowledge) and the real knowledge (Actual Knowledge) of healthcare workers on PPC/PT based on CC in a referral hospital in Northeast Italy.

## 2. Materials and Methods

The study was conducted by a public referral university hospital in Friuli-Venezia Giulia, Italy, which provides healthcare services for children and women, in collaboration with other regional healthcare authorities. On the basis of a regional epidemiological report, between 2018 and 2019, 1188 patients from 0 to 17 years with special healthcare needs were hospitalized for the first time in that facility and 1165 were admitted to other hospitals in the same region. This study is part of a regional project on children with life-limiting or life-threatening conditions, namely “Continuity of care for children with medical complexity: needs and pathways in Friuli-Venezia Giulia region” (2018–2020) and of the “Continuity of Care Project” (2016–2021).

Our questionnaire was designed for health workers (i.e., physicians, nurses and psychologists), working in close contact with pediatric patients, able to access the internet link. Exclusion criterion was refusal to participate. In order to reach potential respondents, as a first dissemination strategy, a formal communication was sent to the Directors of the regional Local Health Authorities (five at the study time), requesting to collaborate and share the survey among their health personnel. Secondly, an e-mail invitation was sent to the members of the working groups of healthcare providers who collaborated with the aforementioned “Continuity of Care Project” (n = 74), the professionals who attended a regional educational event on PPC/PT in 2019 (n = 209) and the psychologists of the Regional Network of Palliative Care (n = 12). Our invitation included a brief information text about the study, a request for participation and the survey link.

A multidisciplinary group of experts, including two physicians, two nurses, two psychologists with experience on PPC and continuity of care for CWSHCN and one health professional education expert contributed to the questionnaire development. The basis for the instrument development was the CC, which identifies eight common areas of competence in the training curricula of nurses, physicians and psychologists: (1) To evaluate the child and family for the access to the PPC/PT network, (2) To receive the child and family in the PPC/PT network, (3) To take charge of the child and family in the different settings of the PPC/PT network, (4) To know how to work as a team, (5) To know how to manage self, (6) To know how to activate and manage a reference center for PPC/PT, (7) To know how to train on PPC/PT, and (8) To know how to conduct research on PPC/PT. The CC also lists knowledge, skills and training strategies for each professional in each area, at different levels. In particular, the CC was used to create items investigating *Perceived* and Wished Knowledge. To measure the Actual Knowledge, we extracted 60 multiple choices questions from the written assessments used in the evaluation of learning outcomes of five previous accredited events of medical continuous training on PC/PPC conducted by an Italian regional training provider. Questions were formulated by teachers with adequate expertise on PC/PPC who have participated in those events. The selection of the final relevant questions for the CC competence areas suitable to a multidisciplinary assessment of Actual Knowledge (nine questions) was based on a consensus after independent and group evaluation by the researchers. The selected questions were slightly edited to be adapted for the online format and increase the comprehensiveness and clarity, following educational research guidelines about assessment modalities (i.e., answer possibilities, and type of written questions) [20,21]. The development process of the questionnaire included four sequential steps and is described in Figure 1. The final questionnaire in Italian is presented as Appendix A.

The minimum sample size was set at 100 in advance to ensure a sample-to-item ratio of 5:1, as recommended for factor analysis [22].

The questionnaire was made available on an internet institutional web page from January to July 2019 and was accessible only through the web link provided by the e-mail invitation. The institutional Quality and Accreditation Department (ED) have regularly verified the proper functioning of the system, checked the compilation of the forms and performed the standard data quality control during the data collection period to ensure the adequacy of the questionnaires. This process was carried out in collaboration with the researchers (EZ, and RV). Anonymized data of all completed questionnaires were exported to Excel files and manually explored by two independent researchers to exclude duplicates.

The socio-demographic characteristics of the respondents were described using counts and percentages. Between-groups differences were evaluated with a Chi-square test (or Fisher test, when appropriate). Total scores were obtained for each scale as the sum of the scores reported for each item. That sum ranged 20–100 for Perceived Knowledge, 8–40 for Wished Knowledge and 0–9 for Actual Knowledge. For each group defined by socio-demographic variables, total scores were reported as median and interquartile range. Furthermore, median scores and interquartile ranges were also calculated for each item and scale according to professional profile (physicians, nurses, and psychologists). The Kruskal–Wallis test was used to evaluate the differences between groups. Three respondents reporting “other” as profession were excluded from the latter comparison.

The internal consistency of scales which were meant to be interrelated (Perceived and Wished Knowledge) were assessed by computing the Cronbach’s Alpha coefficient. Internal consistency was considered good whenever Cronbach’s alpha ≥ 0.70 [23]. Correlation matrices of variables belonging to the same factor were inspected to identify potentially redundant variables. Moreover, scales’ independence was assessed by computing correlation matrices between items belonging to different scales. Factor analysis was performed to identify the underlying structure of each scale. As suggested by Kaiser’s rule, factors with eigenvalues > 1 were extracted [24]. Loadings > 0.4 were retained [25]. The significance level was set at 0.05. The statistical analyses were performed using StataCorp. 2021. Stata Statistical Software: Release 17. College Station, TX, USA: StataCorp LLC.

## 3. Results

### 3.1. Sample

One hundred five questionnaires were completed. Table 1 summarizes the socio-demographic characteristics of the participants by profession. Most respondents were women (90, 85.7%), and nurses (56, 53.3%) and almost a half were 46–60 years old (49, 46.7%). Most participants worked with pediatric patients (86, 81.9%), entirely or partially, with significantly different amounts of activity between professional profiles (*p* = 0.027). The primary work setting for most respondents was the hospital (57, 54.3%), with job profiles unevenly distributed across locations (10 years in most cases (74, 70.5%) and there were no significant differences between professional profiles. Regarding PC/PPC prior education, attendance to congresses or seminars was most frequently reported (63, 60%). Sixty-two respondents (59.1%) declared no work experience in PC/PPC. There were no significant differences between professional profiles, neither for prior education in PC/PPC nor for work experience.

### 3.2. Exploratory Factor Analysis (EFA)

Table 2 shows the results of EFA for Perceived, Wished and Actual Knowledge. The average Cronbach’s Alpha coefficient value was 0.95 for Perceived Knowledge and 0.94 for Wished Knowledge. Four factors were selected for Perceived Knowledge, explaining 75% of the total variability. We interpreted these factors as (1) basic concepts, (2) policies and protocols, (3) communication with child and families, and (4) transitional care. Only one factor was extracted for Wished Knowledge, named desired training. For the Actual Knowledge four factors were retained, explaining 61% of the total variability. These factors were interpreted as (1) basic concepts, (2) policies and protocols, (3) end-of-life issues, and (4) family-centered care.

### 3.3. Levels of Perceived, Wished and Actual Knowledge

Table 3 shows the median scores for each item and the total scores by profession for Perceived and Wished Knowledge.

Perceived Knowledge of psychological and social needs of the child and the family at the end of life was significantly higher for psychologists (*p* = 0.006). Compared to the other professionals, psychologists showed a greater self-perception of their competences also on the topic of mourning (*p* = 0.003). No statistically significant differences were found between profession in terms of Wished Knowledge.

The results for the *Actual Knowledge* were reported in Table 4.

In general, the number of correct answers was significantly higher for psychologists (*p* = 0.022). The discrimination power (DP) was high (>0.5) for all items, except one question on continuity of care (DP = 0.1).

Appendix A shows the total scores for each scale by socio-demographic characteristics of respondents. Perceived Knowledge was significantly different according to prior education/prior working experience in the field of CP/CPP (*p* < 0.001, and *p* < 0.001, respectively). Concerning Wished Knowledge, respondents working primarily within hospital or community showed higher scores compared to those working on both settings (*p* = 0.043). Moreover, different levels of Wished Knowledge were reported based on the length of service experience, with highest scores for senior respondents (*p* = 0.027).

Appendix A shows the proportions of scores ≥ 4 given at each item of the Wished Knowledge scale according to the professional profile (nurse vs. physician vs. psychologist). Overall, team work was the item of major interest (65.7%), followed by the evaluation for PPC/PT network access (61.8%).

## 4. Discussion

This study collected data from different professional profiles and highlighted that psychologists have higher Perceived and Actual Knowledge on PPC/PT; nevertheless, unexpectedly, Wished Knowledge was not significantly different among professions. Findings suggest that the questionnaire has very good internal consistency, as well as a clear underlying structure of the data. These relevant psychometric properties allow for its use in supporting future multi-professional training programs design and/or its implementation in similar settings, alongside other essential strategies for improving health outcomes on pediatric population with palliative care needs.

To our knowledge, there is no previous study on other multi-professional tools directly based on CC that can be compared with our study. Although many surveys have investigated the educational needs of healthcare providers about PPC, few have proposed or used validated tools considering both subjective and objective perspective and/or within a multidisciplinary approach in a European setting. For example, the End-of-Life Professional Caregiver Survey (EPCS) is a validated tool to assess multidisciplinary educational needs based only on a subjective perspective, which was developed and tested for healthcare providers working with adults and pediatric patients in the United States [26,27,28].

Our results show some similarities with these studies. Indeed, psychologists showed higher level of Actual Knowledge about mourning and psychosocial needs of families, compared to nurses and physicians. Similarly, in the study by Lazenby et al., social workers reported more confidence than nurses and physicians about grief counseling for families and generally about cultural and ethical values in palliative and EOL care [27]. We can argue that these issues are professional-specific and this finding could suggest the ability of our tool to discriminate between different professional profiles. Also, sharing “PPC principles and national guidelines” emerged as an educational need of healthcare providers in Schulman-Green et al. as well as “policies and protocols” in our study, where participants showed lower levels of Perceived and Actual Knowledge about these topics [26,27,28].

Concerning the objective evaluation of competences, the Pediatric Palliative Care Questionnaire (PPCQ) included some questions that turned out to be inadequate for drawing conclusions on the tool validity or reliability [17]. Conversely, in our work, all items of Actual Knowledge scale show good discrimination powers, except for one question on continuity of care characterized by 97% of correct answers. This was revised and edited for further use.

Consistent with other studies, in which respondents reported less confidence on “knowing and accessing community PC resources”, our participants have less self-perceived knowledge of transitional cares and are particularly interested in the evaluation of patients for the PPC/PT network [7]. On the other hand, respondents were more comfortable with pain assessment strategies, in line with other findings [29]. It is known that there can be disparities between recognized and unrecognized learning needs, with technical skills vs. interpersonal and intrapersonal skills predominating, respectively [30]. Accordingly, our participants showed higher confidence on communication and psychosocial aspects of PPC than measured with the Actual Knowledge scale. On the other hand, disparities are reduced for Knowledge of policies and protocols. These differences highlight the importance of taking into account both subjective and objective perspective in assessing Knowledge and educational needs. Regarding prior education and/or work experience in PC/PPC, an association with higher Perceived Knowledge was found, in accordance with other studies [26,27,28,29].

Moreover, Wished Knowledge was higher for senior workers. In our opinion, this finding could be justified by the need of continuous training of healthcare professionals working for long time, asking for update and educating them about the emerging topic of PC/PPC. Wished Knowledge levels are also higher for those working primarily in the hospital or in the community compared to those working on both settings. This result may be explained by the fact that professionals working on both settings are more likely to be part of PPC/PC local/regional network, rather than healthcare providers working directly on a single site (hospital vs. community). In our assessment of Wished Knowledge, the most interesting topic for participants was team work, which has also been identified as an educational need of health professionals by other studies [7,29,30]. Based on this result, we believe team work needs to be addressed as specific learning need while training on PPC/PT.

The methodology used for the questionnaire development had several strengths. The four sequential steps start from a recognized national recommendation, i.e., the CC validated by the scientific community, and the involvement of a multidisciplinary health professional group in all rounds of revisions and optimizations assures it content validity. The inclusion of a Wished Knowledge scale implies that health workers’ learning needs may be extensively evaluated by length of service experience and/or working setting before PPC/PT training programs implementation, increasing the utility of this tool for decision makers and educational planners. Indeed, our results for this scale might be useful for decision makers on planning future educational programs. In order to plan effective educational programs, it is necessary to define learning outcomes on the basis of learners’ educational needs, considering prior Knowledge and also experience [31,32,33].

Future research should further explore the acceptability of this tool and barriers faced in the implementation of a similar study without the important support of authorities or institutions.

Our questionnaire represents a valuable tool to assess the educational needs for PPC in a multidisciplinary sample and to evaluate progresses after educational interventions. To date, the establishment of a regional PPC/PT center at the hospital promoting the study has changed the initial context. The center carries out clinical activities and guarantees support, clinical supervision and advice for home care activities throughout the region; among other activities, it develops training and education programs on the subject of PPC/PT. The data collected with this study may be taken into account to address the contents of training courses and the educational interventions on PPC for healthcare providers in the study area. Because of the changes in the reference context, further research could investigate the reliability and discrimination ability of the instrument on a larger sample (e.g., at regional or national level) and test it as pre and post intervention measure to assess the impact of educational interventions and training programs.

As for the implications for practice and research, our study represents a first step to investigate the level of Knowledge (Perceived, Wished and Actual) on PPC/PT of healthcare workers with an easy-to-use instrument, with a multi-professional perspective, based on CC. Indeed, to ensure quality of care, it is necessary to provide education to all professionals involved in the care of children with PPC needs and their families, assess their competences in order to target educational interventions and to measure their impact. In fact, adequate training programs can lead to significant improvements not only on participants’ attitude and skills on the field of PPC, but also the on the care of children with severe conditions [34].

## 5. Study Limitations

This study has some limitations. Data from this single-center study in Italy are not directly generalizable to other settings and our sample is probably affected by auto-selection bias (health providers particularly interested in PPC/PT may have participated). It is difficult to estimate how this selection may have influenced study findings. In addition, this study was based on a relatively limited sample. However, small samples are common among dedicated surveys for health professionals because of the length of forms [35,36,37].

## 6. Conclusions

The questionnaire developed and used for this study has good psychometric properties as an assessment tool of health professionals’ knowledge on PPC/PT in a multidisciplinary perspective. Our findings suggest themes to be considered in designing future training programs and the tool may be used to measure improvements pre- and post- educational interventions. Future research should investigate the acceptability of this tool in other settings and analyze the facilitators and barriers for its implementation.

## Figures and Tables

**Figure 1 healthcare-11-01971-f001:**
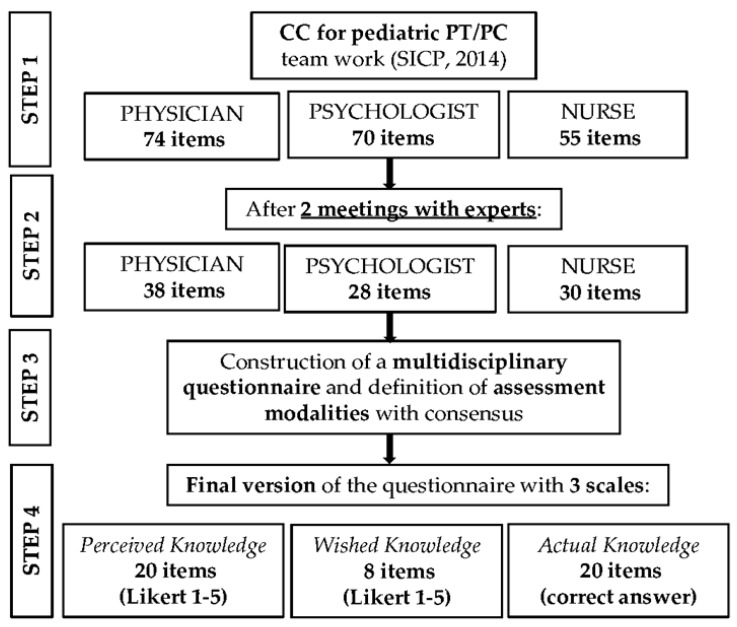
Flow diagram showing the steps of the questionnaire development process. The underline emphasizes the involvement of experts in the item selection process.

**Table 1 healthcare-11-01971-t001:** Sociodemographic characteristic of the respondents.

Socio-Demographic Variables	N = 105	Physician N = 31	Nurse N = 56	Psychologist N = 15	Other N = 3	*p*-Value
Gender, N (%)						0.294
Male	15 (14.3)	8 (25.0)	6 (10.9)	1 (6.7)	0	
Female	90 (85.7)	24 (75.0)	49 (89.1)	14 (93.3)	3 (100)	
Age class, N (%)						0.236
≤30	12 (11.4)	1 (3.1)	8 (14.6)	3 (20.0)	0	
31–45	36 (34.3)	12 (37.5)	18 (32.7)	6 (40.0)	0	
46–60	49 (46.7)	14 (43.8)	27 (49.1)	5 (33.3)	3 (100.0)	
≥61	8 (7.6)	5 (14.6)	2 (3.6)	1 (6.7)	0	
Amount of work with pediatric patients, N (%)						0.027 **
None	19 (18.1)	1 (3.1)	12 (21.8)	5 (33.3)	0	
<50%	15 (14.3)	3 (9.4)	10 (18.2)	2 (13.3)	0	
>50% and <100%	17 (16.2)	4 (12.5)	8 (14.6)	4 (26.7)	1 (33.3)	
100%	54 (51.4)	24 (75.0)	25 (45.4)	4 (26.7)	2 (66.7)	
Primary work setting, N (%)						<0.001 **
Hospital	57 (54.3)	16 (50.0)	34 (61.8)	6 (40.0)	1 (33.3)	
Community	21 (20.0)	0	15 (27.3)	4 (26.7)	2 (66.7)	
Hospital and Community	6 (5.7)	1 (3.1)	2 (3.6)	3 (20.0)	0	
Other	21 (20.0)	15 (46.9)	4 (7.3)	2 (13.3)	0	
Length of service experience, N (%)						0.299
<5 years	19 (18.1)	6 (18.7)	4 (7.3)	2 (13.3)	0	
5–10 years	12 (11.4)	6 (18.7)	8 (14.5)	5 (33.3)	0	
>10 years	74 (70.5)	20 (62.6)	43 (78.2)	8 (53.4)	3 (100.0)	
Prior education on PC/PPC, N (%)						0.675
None	35 (33.3)	13 (40.6)	16 (29.1)	4 (26.7)	2 (66.7)	
Congresses/seminars	63 (60.0)	18 (56.3)	35 (63.7)	9 (59.9)	1 (33.3)	
Advanced courses in PC or PPC (with or without congresses/seminars)	3 (2.9)	0	2 (3.6)	1 (6.7)	0	
Master in PC (with or without congresses/seminars/advanced courses)	4 (3.8)	1 (3.1)	2 (3.6)	1 (6.7)	0	
Prior work experience in PC/PPC, N (%)						0.096 *
None	62 (59.1)	2 (6.2)	3 (5.4)	5 (33.3))	0	
Only PC	18 (17.1)	20 (62.5)	33 (60.0)	6 (40.1)	3 (100.0)	
Only PPC	15 (14.3)	3 (9.4)	13 (23.6)	2 (13.3)	0	
Both	10 (9.5)	7 (21.9)	6 (10.9)	2 (13.3)	0	

Note: PC = palliative care; PPC = pediatric palliative care; ** *p* < 0.05; * *p* < 0.10.

**Table 2 healthcare-11-01971-t002:** Results of the factor analysis.

Scale/Item	Score	Factors			
Perceived Knowledge	Median (IQR)	Basic Concepts	Policies and Protocols	Communication with Child and Families	Transitional Care
Clinical and healthcare needs in PPC/PT	3 (2–3)	0.40			
QOL in PPC/PT	3 (2–4)	0.55			
Personalized care and QOL	3 (2–3)	0.56			
Psychosocial needs in the EOL	3 (2–3)	0.67			
Clinical issues in the EOL	3 (2–3)	0.50			
Ethical dilemmas	3 (2–3)	0.85			
Mourning	3 (2–4)	0.96			
PPC/PT Definition/Philosophy	3 (2–3)		0.75		
PPC/PT Regulatory Framework	2 (1–3)		0.68		
PPC/PT Ethics/Deontology	2 (2–3)		0.60		
PPC/PT Eligibility criteria	2 (1–3)		0.47		
Pain assessment	4 (3–4)		0.70		
PT/PPC Healthcare pathways	2 (2–3)		0.65		
Pain management	3 (2–4)		0.62		
Child’s information right	3 (2–4)			0.68	
Communication with child	3 (2–4)			0.99	
Communication with family	3 (2–4)			0.81	
Child/family’s needs assessment	3 (2–4)			0.83	
Continuity of care	3 (2–4)				0.81
Children–adult services transition	2 (2–3)				0.90
Eigenvalues		10.76	1.76	1.35	1.11
Cronbach’s Alpha by Factor		0.90	0.93	0.89	0.75
Average Cronbach’s Alpha	0.95				
**Wished Knowledge**	**Median (IQR)**	**Desired training**	** *-* **	** *-* **	** *-* **
Evaluation for PPC/PT network access	4 (3–4)	0.86			
Reception in PPC/PT network	4 (3–5)	0.91			
Charge in PPC/PT network settings	4 (3–4)	0.89			
Team work	4 (3–5)	0.84			
Management of self	4 (3–5)	0.83			
PPC/PT Reference center management	3.5 (3–4)	0.80			
Training on PPC/PT	4 (3–5)	0.84			
Research on PPC/PT	3 (2–4)	0.86			
Eigenvalues		5.87			
Cronbach’s Alpha of the Factor		0.94			
**Actual Knowledge**	**N (%)**	**Basic** **concepts**	**Policies and protocols**	**EOL** **issues**	**Family- centered Care**
Pathologies for PPC/PT	48 (47.1)	0.53			
Child’s information right	81 (79.4)	0.87			
Continuity of care	99 (97.1)	0.64			
PPC/PT Regulatory framework	69 (67.7)		0.64		
Buckman Protocol	62 (60.8)		0.71		
Palliative vs. terminal sedation	70 (68.3)		0.62		
Mourning	60 (58.8)			0.71	
EOL symptoms	58 (56.9)			0.62	
Family’s multi-professional care	86 (84.3)				0.84
Eigenvalues		1.56	1.41	1.37	1.18

Note: EOL = end of life, QOL = quality of life, PPC = pediatric palliative care, PT = pain therapy.

**Table 3 healthcare-11-01971-t003:** Median scores by profession: Perceived Knowledge and Wished Knowledge.

Scale/Item Text		Total	Physicians	Nurses	Psychologists	
Perceived Knowledge	Factor	Median (IQR)				*p*-Value
Clinical and healthcare needs in PT/PPC	Basic concepts	3 (2–3)	2.5 (2–3)	2 (2–3)	3 (1–4)	0.430
QOL in PT/PPC	Basic concepts	3 (2–4)	3 (2–4)	3 (2–3)	4 (2–5)	0.116
Personalized care and QOL	Basic concepts	3 (2–3)	3 (2–3)	3 (2–3)	4 (2–5)	0.248
Psychosocial needs in the EOL	Basic concepts	3 (2–3)	2.5 (2.3)	2 (2–3)	4 (2–4)	0.006 **
Clinical issues in the EOL	Basic concepts	3 (2–3)	2.5 (2–3)	3 (2–4)	3 (2–4)	0.342
Ethical dilemmas	Basic concepts	3 (2–3)	3 (2–3)	3 (2–3)	3 (2–4)	0.379
Mourning	Basic concepts	3 (2–4)	2.5 (1.0)	2.7 (1.2)	3.7 (1.0)	0.003 **
PPC/PT Definition/Philosophy	Policies and protocols	3 (2–3)	3 (2–3)	3 (2–3)	3 (2–3)	0.596
PPC/PT Regulatory framework	Policies and protocols	2 (1–3)	2 (1.5–3)	2 (1–3)	3 (1–4)	0.315
PPC/PT Ethics/Deontology	Policies and protocols	2 (2–3)	2.5 (2–3)	2 (2–3)	3 (2–4)	0.049 **
PPC/PT eligibility criteria	Policies and protocols	2 (1–3)	2 (1–3)	2 (1–3)	3 (1–3)	0.355
Pain assessment	Policies and protocols	4 (3–4)	3.5 (3–4)	4 (3–4)	3 (2–4)	0.273
PPC/PT Healthcare pathways	Policies and protocols	2 (2–3)	2 (2–3)	3 (1–3)	2 (2–4)	0.658
Pain management	Policies and protocols	3 (2–4)	2 (3–4)	3 (3–4)	3 (1–4)	0.294
Child’s information right	Communication with child and family	3 (2–4)	3 (2–4)	3 (2–3)	4 (2–5)	0.140
Communication with child	Communication with child and family	3 (2–4)	3 (2–4)	3 (2–4)	4 (2–4)	0.293
Communication with family	Communication with child and family	3 (2–4)	3 (2–4)	3 (2–4)	4 (3–5)	0.171
Child/family’s needs assessment	Communication with child and family	3 (2–4)	3 (2–3)	3 (2–3)	4 (2–5)	0.189
Continuity of care	Transitional Care	3 (2–4)	3 (2–3)	3 (2–4)	3 (2–4)	0.806
Children–adult services transition	Transitional Care	2 (2–3)	2 (2–3)	2 (1–3)	2 (2–3)	0.976
Total score		52 (44–66)	54 (42–60)	51 (44–65)	67 (35–80)	0.341
**Wished Knowledge**	**Factor**	**Median (IQR)**				***p*-value**
Evaluation for PPC/PT network access	Desired training	4 (3–4)	4 (3–4)	4 (3–5)	4 (3–4)	0.460
Reception in the PPC/PT network	Desired training	4 (3–5)	3 (3–4)	4 (3–5)	4 (3–5)	0.364
Charge in the PPC/PT network settings	Desired training	4 (3–4)	4 (3–4)	4 (3–5)	4 (3–5)	0.388
Team work	Desired training	4 (3–5)	4 (3–5)	4 (3–5)	4 (3–5)	0.904
Management of self	Desired training	4 (3–5)	3.5 (3–4)	4 (3–5)	4 (2–5)	0.943
PPC/PT Reference center management	Desired training	3.5 (3–4)	3 (2–4)	4 (3–5)	4 (3–5)	0.117
Training on PPC/PT	Desired training	4 (3–5)	3 (3–4)	4 (3–5)	3 (2–4)	0.155
Research on PPC/PT	Desired training	3 (2–4)	3 (2–3)	4 (2–4)	3 (2–4)	0.150
Total score		28.5 (24–34)	27 (23.5–31.5)	31 (25–37)	27 (23–34)	0.247

Note: EOL = end of life, QOL = quality of life, PPC = pediatric palliative care, PT = pain therapy; ** *p* < 0.05.

**Table 4 healthcare-11-01971-t004:** Number of correct answers by professional profile: Actual Knowledge.

			Total	Physicians	Nurses	Psychologists	
Actual Knowledge	Factor	DP	N (%)				*p*-Value
Pathologies for PPC/PT	Basic concepts	1.00	48 (47.1)	15 (46.9)	20 (36.4)	13 (86.7)	0.002 *
Child’s information right	Basic concepts	0.65	81 (79.4)	24 (75.0)	43 (78.2)	14 (93.3)	0.327
Continuity of care	Basic concepts	0.11	99 (97.1)	31 (96.9)	53 (96.4)	15 (100.0)	1
PPC/PT Regulatory framework	Policies and protocols	0.88	69 (67.7)	18 (56.3)	39 (70.9)	12 (80.0)	0.242
Buckman protocol	Policies and protocols	0.95	62 (60.8)	17 (53.1)	34 (61.8)	11 (73.3)	0.425
Palliative vs. terminal sedation	Policies and protocols	0.86	70 (68.3)	24 (75.0)	35 (63.6)	11 (73.3)	0.51
Mourning	EOL issues	0.97	60 (58.8)	22 (68.8)	32 (58.2)	6 (40.0)	0.178
EOL symptoms	EOL issues	0.98	58 (56.9)	18 (56.3)	32 (58.2)	8 (53.3)	0.962
Family’s multi-professional care	Family-centered care	0.53	86 (84.3)	23 (71.9)	51 (92.7)	12 (80.0)	0.023 **
Total score			6 (5–7)	6 (4–7)	6 (5–7)	7 (6–8)	0.020 **

Note: PPC = pediatric palliative care, PT = pain therapy, DP = discrimination power, ** *p* < 0.05, * *p* < 0.10.

## Data Availability

The datasets used and/or analyzed during the current study are available from the corresponding author on reasonable request.

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
