# Peer review of "Measuring Knowledge of Healthcare Providers on Pediatric Palliative Care with an Online Questionnaire Based on the National Core Curriculum in Italy"

_healthcare, 2023, doi:10.3390/healthcare11131971_

Round 1

Reviewer 1 Report

Dear Authors,

This is an interesting and useful research, and I would recommend publication. However, it needs a bit more detail, either in the text or in longer footnotes, so please answer of all of my questions.

1.       Authors of this paper did not give information about healthcare professionals’ population (physicians, nurses and psychologists), who are employed in pediatric palliative care units in Italy. There is a lack of explanation too the educations needs of healthcare professionals  in the area of pediatric palliative care and therapy of pain, that’s why please complete his information in introduction of this paper.

2.       Please include the information about the opinions of healthcare professionals from different European counties in order to compare the study results. I recommend the article:

https://www.jpsmjournal.com/action/showPdf?pii=S0885-3924%2821%2900657-6

3.       If this study was conducted among 105 healthcare professionals (physicians, nurses and psychologists), so the number of this respondents is low. You should explain precisely, why make you a decision to send survey per internet and why you did not suggested others possibilities in order to increase the number of the respondents?

4.       As You explain in section “Materials and Methods” the healthcare professionals had an access into this survey between January and July 2019, so why you did not make decision to continue this study in order to collect more quantity of survey?  

5.       Please explain all of the used abbreviations in separate section of this paper.

6.       Please introduce the section “study limitations”

Yours faithfully

Reviewer  

Author Response

Dear Reviewer,

Reviewer(s)' Comments to Author:

Reviewer: 1

Dear Authors,

- This is an interesting and useful research, and I would recommend publication. However, it needs a bit more detail, either in the text or in longer footnotes, so please answer of all of my questions.

***Thank you very much for your appreciation and feedback.

- Authors of this paper did not give information about healthcare professionals’ population (physicians, nurses and psychologists), who are employed in pediatric palliative care units in Italy. There is a lack of explanation too the educations needs of healthcare professionals in the area of pediatric palliative care and therapy of pain, that’s why please complete his information in introduction of this paper.

***Thanks for this point. We provided more context in the introduction section by adding details on the different levels of education and skills of healthcare workers in the PC setting (58-62). Moreover, we stressed the importance of implementing adequate training programs on PPC by highlighting an interesting result from a survey carried out by Benini et al. (lines 68-71).

Benini F., Cauzzo C. Training in pediatric palliative care in Italy: still much to do. Ann Ist Super Sanita. 2019 Jul-Sep;55(3):240-245. doi: 10.4415/ANN_19_03_07.

-  Please include the information about the opinions of healthcare professionals from different European counties in order to compare the study results. I recommend the article:

https://www.jpsmjournal.com/action/showPdf?pii=S0885-3924%2821%2900657-6

*** Thanks for this suggestion. We agree that findings of the suggested article are very interesting. For this reason, we have revised and edited the discussion adding this reference.

-  If this study was conducted among 105 healthcare professionals (physicians, nurses and psychologists), so the number of this respondents is low. You should explain precisely, why make you a decision to send survey per internet and why you did not suggested others possibilities in order to increase the number of the respondents?

*** Thanks for this point, which we believe is very relevant and useful as lesson learned for future research. Friuli Venezia Giulia is a Northeast region in Italy where healthcare services for children and women are not uniformly distributed. The dissemination of the survey using internet was decided by the regional project coordination in order to increase recruitment using platforms already used for other project activities. Indeed, we send e-mail invitations and reminders to the Directions of the regional Local Health Authorities as explained in the methods section.

-  As You explain in section “Materials and Methods” the healthcare professionals had an access into this survey between January and July 2019, so why you did not make decision to continue this study in order to collect more quantity of survey?  

*** Thank you for raising this point which gives us the opportunity of giving more information about our study. Data collection period was in line with expected timelines for the regional project about children with life-limiting or life-threatening conditions, namely “Continuity of care for children with medical complexity: needs and pathways in Friuli-Venezia Giulia region” (2018-2020). So, it could not be extended for this study.

- Please explain all of the used abbreviations in separate section of this paper.

*** Thank you for this suggestion. We provided a further section in which all the abbreviations used in the text are listed and explained.

- Please introduce the section “study limitations”

*** Thank you for this point. We added a section we explained in detail the limitation of this study

For more details, please see the revised manuscript.

Reviewer 2 Report

Congratulations to the authors for their significant contribution in developing an online questionnaire to assess healthcare professionals' knowledge of Pediatric Palliative Care (PPC) and Pain Therapy (PT). By basing the questionnaire on the Italian Society for Palliative Care PPC Core Curriculum (CC), the authors have established a solid foundation for assessing healthcare workers' Perceived, Wished, and Actual Knowledge in this specialized area. Notably, the questionnaire exhibited high internal consistency for the Perceived and Wished Knowledge scales, highlighting its reliability. The findings also revealed interesting insights, such as psychologists reporting higher levels of self-perceived skills in specific areas related to PPC/PT, and their ability to provide the highest number of correct answers when tested on Actual Knowledge. Overall, this manuscript represents a significant step forward in assessing healthcare professionals' knowledge in PPC/PT and has the potential to contribute to improved care practices and outcomes for pediatric patients and their families.

Abstract:

The study's objective is clearly stated: to develop an online questionnaire to assess healthcare professionals' perceived, wished, and actual knowledge of PPC/PT. This provides a clear direction for the research. Suggestions:

Consider adding a sentence explaining the significance or potential implications of assessing healthcare professionals' knowledge in PPC/PT.

Provide more details about the field testing process, such as the sample size, selection criteria, and any specific methods used to assess validity and internal consistency. Specify the number of respondents included in the analysis to provide a better understanding of the sample size.

Consider providing some examples or specific areas where psychologists demonstrated higher knowledge, as this would add more depth to the findings.

Elaborate on the potential implications of the study's findings. How can the identified knowledge gaps be addressed, and how might this impact patient care and outcomes?

The introduction provides a general overview of the topic, including the growth of children with special health care needs (CWSHCN) and the need for Palliative Care (PC). It also highlights the lack of knowledge among healthcare workers and the importance of education and training in this field. However, some areas can be improved:

The introduction mentions the Core Curriculum (CC) developed by the Italian Society of Palliative Care but does not provide enough context or details about its content and purpose. Consider briefly summarising the CC and its relevance to the study.

Consider restructuring the introduction to flow more logically and smoothly. The information provided appears fragmented and could benefit from a more precise progression of ideas.

The methods section provides information on participant recruitment, questionnaire development, sample size determination, data collection process, and statistical analyses. However, there is still room for improvement in providing more specific details, such as the criteria for question selection.

Overall, the results section provides a detailed description of the sample characteristics, factor analysis results, and comparisons of knowledge levels among different professional profiles. Including supplementary tables and figures further enhances the presentation of the findings.

Overall, the results obtained support the conclusions drawn in the study. The authors have effectively analyzed the data and provided sufficient evidence to support their findings and interpretations. Aspects to reinforce the validity/interest of your conclusions: Decision-makers and educational planners can utilize the results of this study to plan effective educational programs in PPC. By defining learning outcomes based on healthcare providers' educational needs, prior knowledge, and experience, training initiatives can be tailored to target specific areas of improvement and ensure the delivery of high-quality palliative care. Also,  the study underscores the importance of assessing subjective and objective perspectives in evaluating healthcare professionals' knowledge and educational needs in PPC. Training programs can effectively address disparities and provide comprehensive education covering both technical and interpersonal aspects of palliative care by considering self-perceived knowledge alongside objective assessments.

Author Response

Dear Reviewer,

Reviewer: 2
Comments to the Author:

Congratulations to the authors for their significant contribution in developing an online questionnaire to assess healthcare professionals' knowledge of Pediatric Palliative Care (PPC) and Pain Therapy (PT). By basing the questionnaire on the Italian Society for Palliative Care PPC Core Curriculum (CC), the authors have established a solid foundation for assessing healthcare workers' Perceived, Wished, and Actual Knowledge in this specialized area. Notably, the questionnaire exhibited high internal consistency for the Perceived and Wished Knowledge scales, highlighting its reliability. The findings also revealed interesting insights, such as psychologists reporting higher levels of self-perceived skills in specific areas related to PPC/PT, and their ability to provide the highest number of correct answers when tested on Actual Knowledge. Overall, this manuscript represents a significant step forward in assessing healthcare professionals' knowledge in PPC/PT and has the potential to contribute to improved care practices and outcomes for pediatric patients and their families.

*** Thank you for you appreciation.

-The study's objective is clearly stated: to develop an online questionnaire to assess healthcare professionals' perceived, wished, and actual knowledge of PPC/PT. This provides a clear direction for the research.

*** Thank you for you appreciation.

-Consider adding a sentence explaining the significance or potential implications of assessing healthcare professionals' knowledge in PPC/PT.

***Thank you for this point. We added a comment at the end of the discussion section (lines 308-310), with a reference to another interesting work to motivate the importance of this type of assessment.

Postier C. A., Wolfe J. Education in Palliative and End-of-Life Care-Pediatrics: Curriculum Use and Dissemination. 2022 Mar;63(3):349-358. doi: 10.1016/j.jpainsymman.2021.11.017. Epub 2021 Dec 8. PMID: 34896279.

-Provide more details about the field testing process, such as the sample size, selection criteria, and any specific methods used to assess validity and internal consistency. Specify the number of respondents included in the analysis to provide a better understanding of the sample size.

-Consider providing some examples or specific areas where psychologists demonstrated higher knowledge, as this would add more depth to the findings.

-Elaborate on the potential implications of the study's findings. How can the identified knowledge gaps be addressed, and how might this impact patient care and outcomes?

***Thank for this point. We have revised and edited the discussion section. In particular, we have added a reference to another recent study demonstrating that adequate education and training is crucial to increase workers’ skills and improve the clinical management of patients with complex conditions.

Postier C. A., Wolfe J. Education in Palliative and End-of-Life Care-Pediatrics: Curriculum Use and Dissemination. 2022 Mar;63(3):349-358. doi: 10.1016/j.jpainsymman.2021.11.017. Epub 2021 Dec 8. PMID: 34896279.

-The introduction provides a general overview of the topic, including the growth of children with special health care needs (CWSHCN) and the need for Palliative Care (PC). It also highlights the lack of knowledge among healthcare workers and the importance of education and training in this field. However, some areas can be improved:

***Thank you for your appreciation. The introduction was revised and edited according to referees’ suggestions.

-The introduction mentions the Core Curriculum (CC) developed by the Italian Society of Palliative Care but does not provide enough context or details about its content and purpose. Consider briefly summarising the CC and its relevance to the study.

***Thank you for this suggestion. We revised and edited the introduction providing further details on what is the CC (lines 55-62).

-Consider restructuring the introduction to flow more logically and smoothly. The information provided appears fragmented and could benefit from a more precise progression of ideas.

***Thank you for this point. We revised and edited the introduction. In particular, we have made slight changes in the order of presentation of our arguments to increase clarity for readers.

-The methods section provides information on participant recruitment, questionnaire development, sample size determination, data collection process, and statistical analyses. However, there is still room for improvement in providing more specific details, such as the criteria for question selection.

***Thank you for this suggestion. We revised and edited the methods section providing further details on this aspect: questions were selected by consensus among experienced researchers by independent and group evaluation.

-Overall, the results section provides a detailed description of the sample characteristics, factor analysis results, and comparisons of knowledge levels among different professional profiles. Including supplementary tables and figures further enhances the presentation of the findings.

***Thank you for you appreciation.

-Overall, the results obtained support the conclusions drawn in the study. The authors have effectively analyzed the data and provided sufficient evidence to support their findings and interpretations. Aspects to reinforce the validity/interest of your conclusions: Decision-makers and educational planners can utilize the results of this study to plan effective educational programs in PPC. By defining learning outcomes based on healthcare providers' educational needs, prior knowledge, and experience, training initiatives can be tailored to target specific areas of improvement and ensure the delivery of high-quality palliative care. Also,  the study underscores the importance of assessing subjective and objective perspectives in evaluating healthcare professionals' knowledge and educational needs in PPC. Training programs can effectively address disparities and provide comprehensive education covering both technical and interpersonal aspects of palliative care by considering self-perceived knowledge alongside objective assessments.

***Thank you for you appreciation.

For more details, please see the revised manuscript.